# Interspecific synchrony on breeding performance and the role of anthropogenic food subsidies

Ana Payo-Payo [1,2]*, José-Manuel Igual[2,3], Ana Sanz-Aguilar[3,4], Enric Real[3,5], Meritxell Genovart [2,6], Daniel Oro[2,6], Giacomo Tavecchia[3]

**1** School of Biological Sciences, University of Aberdeen, Aberdeen, United Kingdom, **2** IMEDEA (CSIC-UIB), Esporles, Spain, **3** Animal Demography and Ecology Unit (GEDA), IMEDEA (CSIC-UIB), Esporles, Spain, **4** Applied Zoology and Conservation Group, University of Balearic Islands, Palma, Spain, **5** Instituto Español de Oceanografía, Centre Oceanográfico de Baleares, Palma, Spain, **6** CEAB (CSIC), Blanes, Spain

* ana.payo-payo@abdn.ac.uk

**Data Availability Statement:** The data and code are available in the Supporting information files.

**Funding:** This work has been partially supported by Balearic Government, Ministry of Education

## Abstract

Synchrony can have important consequences for long-term metapopulations persistence, community dynamics and ecosystems functioning. While the causes and consequences of intra-specific synchrony on population size and demographic rates have received considerable attention only a few factors that may affect inter-specific synchrony have been described. We formulate the hypothesis that food subsidies can buffer the influence of environmental stochasticity on community dynamics, disrupting and masking originally synchronized systems. To illustrate this hypothesis, we assessed the consequences of European policies implementation affecting subsidy availability on the temporal synchrony of egg volume as a proxy of breeding investment in two sympatric marine top predators with differential subsidy use. We show how 7-year synchrony appears on egg volume fluctuations after subsidy cessation suggesting that food subsidies could disrupt interspecific synchrony. Moreover, cross correlation increased after subsidy cessation and environmental buffering seems to act during synchronization period. We emphasize that subsidies dynamics and waste management provide novel insights on the emergence of synchrony in natural populations.

## Introduction

Synchrony—defined as a pronounced temporal or spatial clustering of biological and ecological processes—has captivated ecologist for decades [1–4]. A considerable number of studies has focused on understanding causes and consequences of intra-specific synchrony for metapopulation dynamics, stability, and persistence [2, 5–7]. Intra-specific synchrony is apparent at large spatial scale from correlated environmental conditions, the so-called 'Moran's effect' [6, 8]. At a small spatial scale, other factors such as dispersal [9], parasitism [10], trophic interactions [11] or predation [12] can drive population synchrony/asynchrony.

(FPU2012-000869), Ministry of Economy (IBISES-CGL2013-42203-R and RESET-CGL2017-85210-P) and EU project MINOW (H2020-634495). It was partially funded by the EU FEDER regional funds. ASA was supported by a Ramón y Cajal contract (RYC-2017- 22796) funded by the Ministerio de Ciencia, Innovación y Universidades, the Agencia Estatal de Investigación and the ESF. MG was supported by a postdoctoral contract co-funded by the Regional Government of the BI and the ESF (PD/023/2015). There was no additional external funding received for this study. The funders had no role in study design, data collection and analysis, decision to publish, or preparation of the manuscript.

**Competing interests:** The authors have declared that no competing interests exist.

Despite synchrony studies having conventionally been carried out on single-species, recent developments in multispecies approaches have provided tools to explore insights into how communities respond to environmental drivers [5, 13, 14]. For instance, climate change-related events are known to alter spatial synchrony of plankton dynamics [15] and to synchronize vertebrate population fluctuations across communities of herbivores [16]. However, inter-specific synchrony in demographic parameters other than population size have received considerably less attention [13, 17]. Breeding parameter—such as egg volume- often reflect conditions in the ecosystems [18]. Therefore, using inter-specific synchrony of breeding parameters could be a useful indicator of local environment state [17].

Here, we formulate a new testable idea explaining inter-specific synchrony disruption in natural systems, the *subsidy-decoupling hypothesis*. Anthropogenic food subsidies shape communities—by altering processes such as competition, predator–prey interactions—and ecosystems worldwide and their impact has been documented from aquatic to terrestrial environments [18]. These subsidies are nowadays an important element in the ecology of many wild species and communities as they can potentially alter the original trophic niche, reduce the demographic variance, or/and buffer the impacts of environmental stochasticity [18, 19]. Because different taxa exploit subsidies in different ways, subsidies are expected to impact synchrony patterns across populations through contrasting effects on their fitness components (e.g., survival and reproductive parameters [19–21]). Since the exploitation of subsidies buffers populations from the impact of environmental stochasticity, it would also decouple the dynamics of sympatric species with differential subsidy use [18]. In such conditions, the removal of food waste would be expected to reveal the underlying synchrony dynamics otherwise hidden.

In the last decaces the European Union (EU) has enforced environmental policies to reduce anthropogenic food subsidies availability and to mitigate their effects on wildlife and human well-being [22]. Such policies include the Landfill Waste Council Directive (LWCD, [23])—aiming for the termination of open landfilling—and the Landing Obligation Directive (LOD, [24])—enacting a ban of fisheries discards. The implementation of these policies is a fantastic opportunity to test our hypothesis since they ultimately aim to modify the availability of anthropogenic food subsidies for wildlife [25, 26]. Here, we focus on the effect of the transformation of Mallorca's single open-landfill (Son Reus) into an incinerator following LWCD on the changes in the egg volume of two sympatric avian top predators. Mallorca's landfill transformation resulted in previously abundant and accessible human food waste becoming inaccessible to scavengers.

One of the species known to, regularly, exploit such subsidies was the Yellow-legged gull (*Larus michahellis*). Yellow-legged gull is a generalist species with a broad dietary spectrum including terrestrial and aquatic prey, fisheries discards and, anthropogenic food waste [19, 21]. Yellow-legged gull landfill waste exploitation varies in extent throughout their distribution range [27]. However, in our focal Yellow-legged gull study colony landfill waste represented up to 60% of their diet [28] until it became inaccessible, forcing gulls to switch diet towards more marine resources [19]. Further, as a result of the landfill closure, our focal Yellow-legged gull population experienced significant decreases in breeding parameters (i.e., clutch size, and egg volume), but there was no significant change in survival probability [19, 21]. Yellow-legged gulls at our studied colony breed sympatrically (~2 Km apart) with another marine top predator, the Scopoli's shearwater (*Calonectris diomedea*).

Scopoli's shearwater is a marine specialist seabird specialized on small epipelagic fish and squid [29]. Scopoli's shearwater can make opportunistic use of other subsidies such as fishing discards but they do not exploit landfill waste [30]. Despite Yellow-legged gull and Scopoli's shearwater have contrasting foraging strategies, they both are income breeders and their egg

volume is a good proxy of food availability just before and during the egg formation period [31, 32]. We hypothesize that while anthropogenic food subsidies from the landfill were available—until 2010—Yellow-legged gull egg volume decoupled from environmental stochasticity and from Scopoli's shearwater dynamics (e.g. [21]). Once anthropogenic food subsidies from the landfill were no longer available, we expect Yellow-legged gull to be affected by the similar environmental stimuli than Scopoli's shearwater, increasing the degree of synchrony. Moreover, we expect food subsidies should buffer egg volume—as a proxy of breeding investment—against environmental variation so that egg volume is larger when environmental stochasticity is removed.

## Material and methods

### Data and study species

We collected egg volume data of Yellow-legged gull and Scopoli's shearwater between 2002 and 2019 at Dragonera Natural Park, a 380-ha reserve off the western coast of Mallorca Island, Spain. Yellow-legged gull breeds in areas of gentle slope and low vegetation, laying 2–3 eggs in March-April. Scopoli's shearwater breeds in burrows under boulders or vegetation, laying single-egg clutches during May. In the fifteen-year period considered here, we measured a total of 1897 eggs of Scopoli's shearwater (1897 nests) and 1743 eggs of Yellow-legged gull from 3-egg clutches (581 nests) to the nearest millimetre using a digital calliper (See S1 Table for data annual sample sizes). Egg volume ($V$), expressed in cm$^3$, was calculated as $V = \beta(L)(W)^2$, where $L$ and $W$ are egg length and width, respectively, and $\beta$ is a species-specific constant [33, 34]. Here $\beta$ is $0.509 \times 10^{-3}$ and $0.476 \times 10^{-3}$ for Scopoli's shearwater and Yellow-legged gull, respectively [33, 34]. We use annual mean of egg volume per clutch as a proxy of breeding investment since it is known to be positively correlated with food availability in both species [19, 31]. This study complies with the current European and Spanish laws regulating scientific research on animals. Permits were given by Spanish Ministry of the Environment and Dragonera Natural Park Authorities at Govern Balear.

### Analysis of synchrony

We analysed our data by two traditionally used and complementary methods: cross-correlation and a State-Based Markov Chain modelling [5]. Cross correlation analyses between time-series are common methods to measure synchrony; however, although it considers the magnitude of the change, it does not necessarily reflect dynamics coupled in time [6]. Consequently, we applied State-Based Markov Chain modelling [5].

Following Haydon *et al.* (2003), we considered two time series $\{X_{i,t}\}$ of $k$ values of populations i (with i = 1,2) at time t (with t = 1, 2,.., $k$). We are interested in the degree of synchrony, that is to say the extent to which fluctuations of the two series are aligned. For each time-series we built a new one $\{Y_t\}$ of $k$-2 elements (with t = 2, 3,..., $k$-1) where:

$Y_t = 1$ when $X_{t-1} > X_t \leq X_{t+1}$ ($X_t$ is a trough in the time series)

$Y_t = 2$ when $X_{t-1} \leq X_t \leq X_{t+1}$ ($X_t$ is the intermediate value of two consecutive increases)

$Y_t = 3$ when $X_{t-1} \leq X_t > X_{t+1}$ ($X_t$ is a peak in the time series)

$Y_t = 4$ when $X_{t-1} > X_t > X_{t+1}$ ($X_t$ is the intermediate value of two consecutive decrease years).

The series $\{Y_t\}$ describes the variable state at a given point in time in relation to the precedent and subsequent states. Considering the new series $\{Y_t\}$ as a first order Markov process, there is a 4x4 matrix, **T**, that describes the transition probabilities, $\tau_{(r, m)}$, from state r to state

m. Let the elements, $s(j)_t$, of a row vector $\mathbf{s}_t$ be the proportion of time series in each state at time t, with $j = 1, 2, \ldots, 4$ and $t = 2, 3, \ldots, k\text{-}1$. As a measure of state synchronization, we adopt the entropy, $H_t$. This measure is equivalent to the Shannon–Weaver diversity indicator (e.g., [35]) estimated as:

$$Ht = -\sum_{j=1}^{4} s(j)_t \ln\big(s(j)_t\big).$$

Let $H_{null}$ denote possible values of $H_t$ under the hypothesis that the two-time series are stationary independent Markov chains, all associated with the transition matrix $\mathbf{T}$. We approximate the distribution of $H_{null}$ by simulation, noting that $\mathbf{s}_{t+1} = \mathbf{s}_t \, \mathbf{T}$, and using $\mathbf{T}$ as the estimate of the transition matrix to simulate the dynamic $\mathbf{s}$ of each of the n population treated as separate and independent as in [5]. A useful measure of degree of synchronization is then the quantity $\Phi_t = 1 - \frac{H_t}{H_{null}}$ (where $H_{null}$ is the expected value of $H_{null}$). At any given time $t$, if the time series are in different states thus desynchronized (e.g. one time series growing, state 2, and one time series decreasing, state 4) $\Phi_t$ will be close to zero. However, if at time $t$ all time-series are in the same state (e.g. both time series growing, state 2) thus fully synchronized $\Phi_t$ will be close to one.

## Environmental buffering

We define environmental buffering as the reduction of the effects of environmental variation on vital rates under certain conditions. For instance, food subsidies should buffer egg volume against environmental variation by providing a more predictable and reliable food source. To test potential environmental buffering, we use North Atlantic Oscillation climatic index during December–March a proxy of environmental conditions (hereafter $W_{NAO}$, [36]). High positive WNAO values are associated with the intensification of upwelling and small pelagic fish availability [36] thus higher food availability for seabirds. We used $W_{NAO}$ because previous studies in the same study area showed that it correlates with Scopoli's shearwater egg volume and annual breeding success [37]. We used generalized linear models (GLM, [38]) to evaluate the effects of landfill closure (i.e. before vs after), species (i.e. Yellow-legged gull vs Scopoli's shearwater), year, synchrony (i.e. $\Phi = 1$ vs $\Phi \neq 1$) and $W_{NAO}$ on mean egg volume. Models were selected using the Akaike Information Criterion corrected for small sample size (AICc value [39]). We considered the model with the lowest AICc as the best model and those within four points of ΔAICc (the difference in AICc) to be equivalent and we used them for model averaging [39]. Further, we consider an effect significant when the CI of the beta did not overlap with zero. Additionally, we assessed if Yellow-legged gull average annual egg volume was more correlated with environmental conditions during years of high synchrony measures ($\Phi == 1$). To do this we coded a new binomial variable, that took value of 1 on the years when synchrony was perfect and 0 the remaining years. All analysis were conducted in R software [40] and code is available in S1 Appendix.

## Results

The cross-correlation (CC) increased after landfill closure from $CC_{before}$ = -0.255 to $CC_{after}$ = 0.765 and state-based Markov chain modelling analysis showed that both species presented significantly synchronous dynamics for seven years after the landfill closure (Fig 1). Our results show that synchrony stopped after 2017. The model selection procedure in modelling egg volume resulted in five models occurring within 4 points of AICc (Table 1 and S2 Table). Egg volume of Yellow-legged gull was significantly higher than Scopoli's shearwater ($\beta_{Lm}$ =

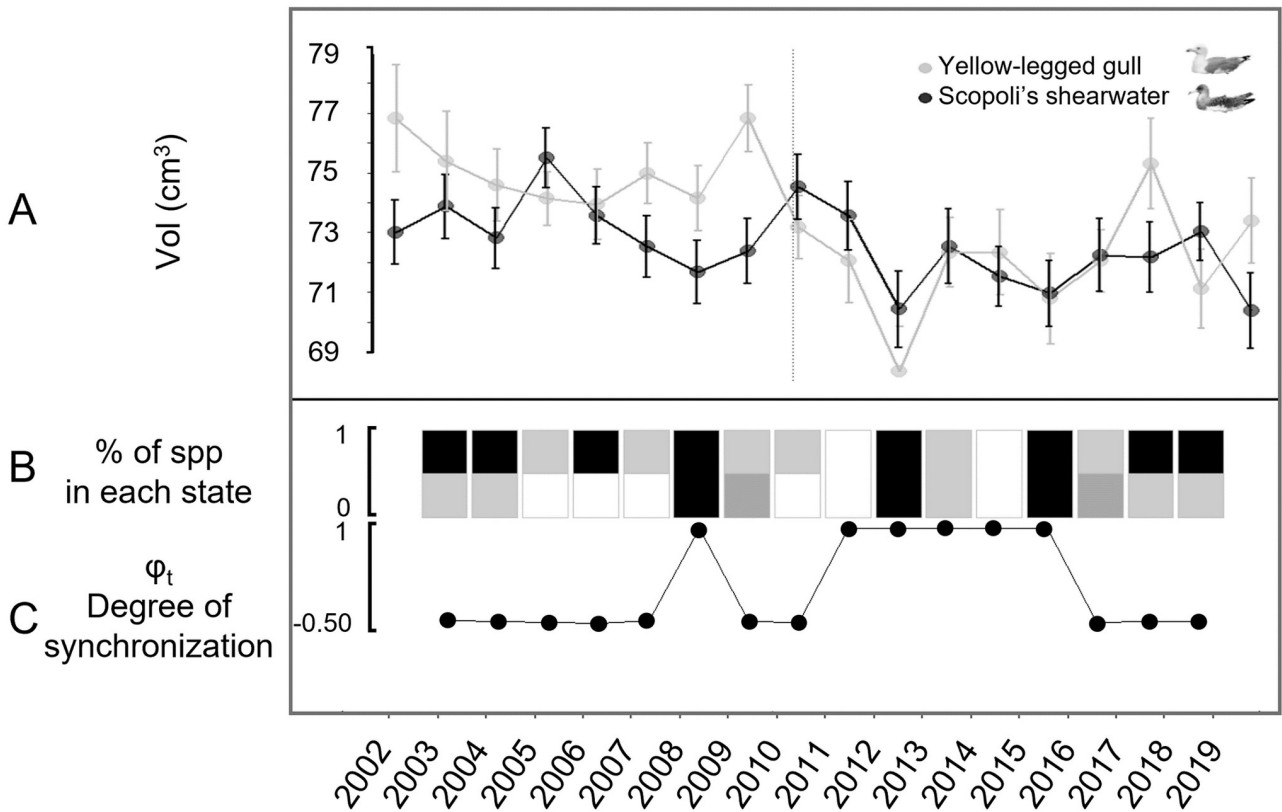

**Fig 1. Scopoli's shearwater and the Yellow-legged mean egg volume synchrony results.** a) Mean egg volume 95% confidence intervals (grey) of the Scopoli's shearwater and the Yellow-legged gull (black) breeding at Dragonera Natural Park (2002–2019, Spain). Vertical dashed line indicates landfill closure. b) Proportion of species in: state1 (trough, black); 2 (increase, grey stripped); 3 (peak, white); and 4 (decrease, grey). c) Synchrony score values ($\Phi_t$), 1 = synchrony dynamics.

1.457 CI = [0.16, 2.75]) except during the synchronization period ($\beta_{LM}$:Sync$_{YES}$ -2.014 CI = [-3.85, -0.18], see S3 Table for further details). $W_{NAO}$ had a significant negative and additive effect on egg volume for both species through the whole study period (-0.388 CI = [-0.69, -0.08], S3 Table). The interaction effects between species, $W_{NAO}$ and synchrony were not significant, but suggested a decrease in environmental buffering of Yellow-legged gull egg volume

**Table 1. Modelling of egg volume of yellow-legged gulls and Scopoli's Shearwaters on Dragonera Natural Park, Spain.**

| Model | df | logLik | AIC$_c$ | ΔAIC$_c$ | ω |
|---|---|---|---|---|---|
| **Sp + Sync + W$_{NAO}$ + Sp:sync** | **6** | **-58.14** | **131.17** | **0.00** | **0.43** |
| Sp + Sync + W$_{NAO}$ + Sp:Sync + Sp: W$_{NAO}$ | 7 | -57.72 | 133.44 | 2.27 | 0.14 |
| Sp + Sync + W$_{NAO}$ | 5 | -60.89 | 133.78 | 2.61 | 0.12 |
| Sp + Sync + W$_{NAO}$ + Sp:Sync + Sync: W$_{NAO}$ | 7 | -58.13 | 134.27 | 3.10 | 0.09 |
| Sync + W$_{NAO}$ | 4 | -62.78 | 134.86 | 3.69 | 0.07 |

The best model is shown in bold. Notations are Sp, species; sync, synchrony; W$_{NAO}$, Winter North Atlantic Oscilation Index; "+", additive effect; ":", interaction effect;df, degrees of freedom; logLik, Log-Likelihood, AICc, Akaike's information criterion corrected for sample size; ΔAICc, AICc difference with the best model; w, weight of the model. See S2 Table for full model selection table.

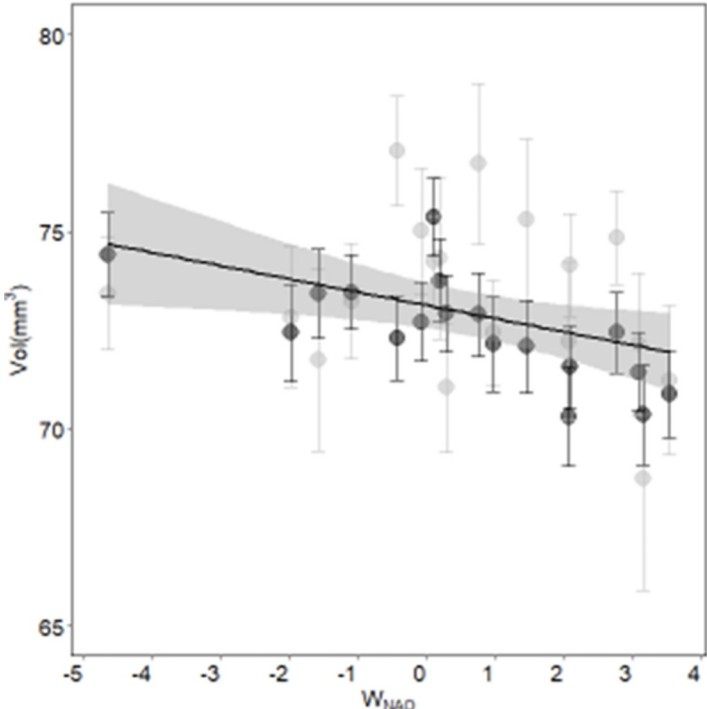

**Fig 2. Correlation ($R^2$ = 0.152) and linear model beta estimate ($\beta W_{NAO}$ = -0.33) between $W_{NAO}$ index and mean annual egg volume and 95% confidence interval.** Scopoli's shearwater (black) and the Yellow-legged (grey) model predictions (black line with 95% confidence grey bands).

following the landfill closure varying from 0% (p = 0.898) during the non-synchronic phase, to 43% (p = 0.110) during the synchronic phase (2010–2016, See Fig 2).

## Discussion

Identifying new drivers of synchrony has deep practical importance since it can boost our opportunities to understand population and community regulations. Here we test for the subsidy-decoupling hypothesis which explains inter-specific synchrony disruption in natural systems. Cross-correlation and State-Based Markov Chain analysis were consistent in suggesting that variation in egg volume respond to similar drivers. The synchrony analysis showed perfect synchronization ($\Phi_t$ = 1) after 2009. Synchronous dynamics lasted for 7 years after the landfill was closed in 2010. In essence, both species egg volume time-series oscillated consistently over time. Further, results indicate that environmental buffering decreased in Yellow-legged gull egg volume dynamics during the synchronized years, yet this was not statistically significant most likely due to low sample size.

After closure of the open-air landfill, and consequent disappearance of a main local source of anthropogenic food subsidies, the synchrony in egg volume between the two sympatric species could potentially arise from higher overlap of both species' trophic niches. Although evidence of increase diet overlap between the studied species is lacking, diet shift could potentially have resulted in increasing interference competition [41, 42]. This could be the result of changes in foraging strategies and behaviour of the species so far relying on the disappeared subsidies and/or to the permanent dispersal of those individuals specialized in consuming human waste resources [19]. In our studied site, after the closure of the open-landfill, Yellow-legged gulls progressively shifted their diet towards marine resources thus compromising

trophic segregation [19]. Yellow-legged gulls breeding at Dragonera Natural Park have previously exhibited mechanisms whereby their behaviour can be adjusted [19, 21]. The 7-year synchrony period could be related to the time associated with acquiring and transferring knowledge about the novel foraging landscape and identifying new foraging grounds [19, 43].

A second explanation for the observed synchronization pattern is that the breeding investment of the two top predators is indirectly influenced through different paths by common drivers potentially, e.g., coupled dynamics of food resources driven by environmental forcing [44]. The breeding success of Scopoli's shearwater for example in our population was negatively associated with winter North Atlantic Oscillation [45], due to increasing marine productivity associated with water mixing [46] or to a carry-over effect of the winter conditions [47]. Breeding parameters (i.e., clutch size and egg volume) of Yellow-legged gulls in our population during the 7-year synchrony period were driven by food waste availability and that they started consuming a more marine diet after the landfill was closed [48]. It is possible that marine prey exploited by Yellow-legged gull and Scopoli's Shearwater are susceptible to similar environmental forcing and results in coupled dynamics.

Finally, a third possible explanation is landfill closure constituted a strong perturbation leading to temporary transient synchrony between Yellow-legged gull and Scopoli's shearwater [49], as a response to the rapid decrease in Yellow-legged gull population size [19, 50]. Transient synchrony dynamics have previously been suggested in invertebrates experiencing seasonal environments and changes in interspecific interactions [50]. However, we argue that the mechanisms from transient dynamic do not explain the disappearance of synchrony after seven years, and that changes in interspecific competition at sea is likely the cause of this pattern. Ultimately, although both, cross-correlation and state-based Markov chain modelling, provided clear evidence of egg volume synchronization in our studied species after human waste from the landfill became unavailable, we can't unequivocally identify the underlaying mechanisms. We acknowledge that longer time series and that further research would certainly shed some clarification to our findings.

Overall, contrary to prior views, we show that anthropogenic food subsidies effects are not restricted to their availability but also to their dynamics (i.e., sudden termination) and that they can mask otherwise naturally synchronized systems. Therefore, the contribution of anthropogenic subsidies to population and community dynamics might be more pervasive than previously thought and deserves further empirical attention. Future research testing the subsidy-decoupling hypothesis might benefit from accidental experimental opportunities such as the implementation of recent environmental policies such as Landing Obligation and Landfill Waste Council Directives or from sudden population collapses resulting from current environmental change [18, 23, 25].

## Supporting information

**S1 Table. Study data summary.**
(DOCX)

**S2 Table. Extended model results.**
(DOCX)

**S3 Table. Model averaging estimates.**
(DOCX)

**S1 Appendix. Code.**
(DOCX)

## Acknowledgments

We would like to thank all people who helped us on the field; Martí Mayol and the staff of the Dragonera NP for helping with logistics, and to Pablo Almaráz and D. Haydon for fruitful discussions.

## Author Contributions

**Conceptualization:** Ana Payo-Payo, Giacomo Tavecchia.

**Data curation:** Ana Payo-Payo, José-Manuel Igual, Giacomo Tavecchia.

**Formal analysis:** Ana Payo-Payo.

**Funding acquisition:** Daniel Oro, Giacomo Tavecchia.

**Investigation:** Ana Payo-Payo.

**Methodology:** Ana Payo-Payo.

**Project administration:** Ana Payo-Payo.

**Supervision:** Ana Sanz-Aguilar, Daniel Oro, Giacomo Tavecchia.

**Visualization:** Ana Payo-Payo.

**Writing – original draft:** Ana Payo-Payo.

**Writing – review & editing:** José-Manuel Igual, Ana Sanz-Aguilar, Enric Real, Meritxell Genovart, Daniel Oro, Giacomo Tavecchia.

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
