## [Decision Letter · Decision Letter 0]

22 Jun 2022

PONE-D-22-11852Interspecific synchrony on breeding performance and the role of anthropogenic food subsidies.PLOS ONE

Dear Dr. Payo Payo,

Thank you for submitting your manuscript to PLOS ONE. After careful consideration, we feel that it has merit but does not fully meet PLOS ONE’s publication criteria as it currently stands. Therefore, we invite you to submit a revised version of the manuscript that addresses the points raised during the review process.

We look forward to receiving your revised manuscript.

Kind regards,

Jose M. Riascos, Ph.D.

Academic Editor

PLOS ONE

Journal Requirements:

- https://www.tdx.cat/handle/10803/668552#page=1

In your revision ensure you cite all your sources (including your own works), and quote or rephrase any duplicated text outside the methods section. Further consideration is dependent on these concerns being addressed.

5. Thank you for stating in your Funding Statement: 

"This work has been partially supported by Balearic Government, Ministry of Education (FPU2012-000869), Ministry of Economy (IBISES-CGL2013-42203-R and RESET-CGL2017-85210-P) and EU project MINOW (H2020-634495). ASA was supported by a Ramón y Cajal contract (RYC-2017- 22796) funded by the Ministerio de Ciencia, Innovación y Universidades, the Agencia Estatal de Investigación and the ESF. MG was supported by a postdoctoral contract co-funded by the Regional Government of the BI and the ESF (PD/023/2015)."

6. Thank you for stating the following financial disclosure: 

"This work has been partially supported by Balearic Government, Ministry of Education (FPU2012-000869), Ministry of Economy (IBISES-CGL2013-42203-R and RESET-CGL2017-85210-P) and EU project MINOW (H2020-634495). ASA was supported by a Ramón y Cajal contract (RYC-2017- 22796) funded by the Ministerio de Ciencia, Innovación y Universidades, the Agencia Estatal de Investigación and the ESF. MG was supported by a postdoctoral contract co-funded by the Regional Government of the BI and the ESF (PD/023/2015)."

7. In your Data Availability statement, you have not specified where the minimal data set underlying the results described in your manuscript can be found. PLOS defines a study's minimal data set as the underlying data used to reach the conclusions drawn in the manuscript and any additional data required to replicate the reported study findings in their entirety. All PLOS journals require that the minimal data set be made fully available. For more information about our data policy, please see http://journals.plos.org/plosone/s/data-availability.

Reviewers' comments:

Reviewer's Responses to Questions

**Comments to the Author**

1. Is the manuscript technically sound, and do the data support the conclusions?

Reviewer #1: Yes

Reviewer #2: Yes

2. Has the statistical analysis been performed appropriately and rigorously? 

Reviewer #1: Yes

Reviewer #2: No

3. Have the authors made all data underlying the findings in their manuscript fully available?

Reviewer #1: Yes

Reviewer #2: Yes

4. Is the manuscript presented in an intelligible fashion and written in standard English?

Reviewer #1: Yes

Reviewer #2: Yes

5. Review Comments to the Author

Reviewer #1: I’d like to thank the authors for creating this interesting article, that takes an original look at the phenomenon of anthropogenic food subsidies in vertebrate populations, adding the community aspect, which is rarely assessed in this context. I enjoyed reading it, it is well-written, and I believe the methods are sound. My main criticism would be that the time series is still too short to confirm certain trends that are suggested in the discussion. On the other hand, we can’t wait for decades to report on demographic monitoring trends when they concern aspects relating to the short and mid-term consequences of policy changes on wildlife. I refer to my detailed comments on the discussion for what I think might be a bit weak in terms of conclusions. I believe this manuscript is fit for publication after minor revision. I look forward to a revised version.

Detailed comments:

L46-67 This general introduction to synchrony, and its focus on intra-specific, inter-population synchrony, is a bit too long and far-reaching. Could the authors try to shorten it, perhaps into a single paragraph?

L68 replace “have” by “having” or “despite that”

L76 ‘considerably’

L78-79 I don't see how one follows the other: synchrony is a useful indicator of environment state because breeding parameters reflect environmental conditions?

L84 “These subsidies”

L89; 95-96 I suggest to avoid an ambiguous use of the words “nature” and “natural”

L92 perhaps replace “species” by “populations”?

L 97 The term "subsidy" has other connotations when used in combination with "EU" (i.e. economic incentives). How about replacing it by "waste" or "food waste"?

L101 “aiming for the termination”

L103 “a ban on fisheries discards”

L104 delete “experimentally”. Also “aim to modify”

L105 “of anthropogenic food subsidies for wildlife” Please be generous with the non-specialized reader

L111 “waste becoming inaccessible”

L124 sympatry: if the nesting grounds do not overlap, do the home ranges overlap? How much does the spatial niche of these species actually overlap?

L152 do we have any evidence for a relationship between maternal investment and breeding success in these species? If not, references to works on similar species may instead be cited

L158 replace ‘despite’ by ‘although’

L159 replace “necessary” by “necessarily”

L172 delete “we used we used the”

L181-183 do you mean by this that you included the interaction between NAO and your proxy for synchrony in the full model? Please state explicitly which interaction were included and to what end

L187; 206; 223 I’m afraid I’m misinterpreting the figure… but it seems like synchrony is observed only in 5 years after landfill closure? But you consistently mention a 7-year synchrony period throughout the manuscript. Can you please clarify this to me in the rebuttal letter?

L209 “decreased”

L210 replace “were” by “was”

L211 Perhaps be a bit more explicit on what this event was: "After closure of the open-air landfill, and consequent disappearance of a main local source of anthropogenic food subsidies, ..."

L213 – 214 ‘the foraging strategy of one species’: I’d rather explicitly say here “of the species so far relying on the disappeared subsidies”. I feel like the more interesting trends are here given by Lm, while the discussion is treating both species equally.

L214 “and/or”

L214-215 do the numbers (nLm and nCd) reported in Table A.1 reflect total population size of the study colonies? Are there discernible trends in these population sizes, particularly in relation to the closure of the landfill?

L223 ‘time associated with acquiring’

L224 so this reported synchrony is interpreted as symptomatic of a decreased efficiency in the use of the surrounding landscape by the Lm's nesting in Dragonera?

L231 so marine productivity increases under more positive NAO conditions? It would be helpful if either in the introduction or earlier in the discussion a short explanation is provided on what NAO is and what it most probably implies from the point of view of breeding seabirds.

L233 “carry-over”

L235 What would be the mechanism linking population size change to synchrony in maternal investment? I fail to understand how the latter would be a consequence of the former without more details. Is it that Lm reproductive individuals exploiting garbage migrate away from the population, with only those Lm individuals that respond in their reproductive performance similarly to Cd remaining in the colony? I feel like abstract terminology is sometimes getting in the way of the interpretation of results in this discussion.

L239 is there any reason to suspect a similar (interspecific interaction) mechanism to be at play in this context? If so, which one exactly? Feeding competition at sea?

L249 "non-linear dynamics": Please be more explicit on exactly what you claim to prove. Are you saying that we have evidence that the sudden disappearance of food subsidies induces in itself a response in population dynamics that would not be expressed if this disappearance had been gradual? What is this claim based on?

L252 Did we previously not think that anthropogenic food subsidies buffer natural variability in population dynamics of the species exploiting them? And that this had consequences on the community dynamics? In the introduction it was expected, based on previous knowledge, that the population so far relying on subsidies would respond to environmental fluctuations more similarly to a population that did not rely on subsidies, because these subsidies decouple population dynamics from non-anthropogenic environmental variation. I would rather argue that the results confirm this already existing theory from a novel point of view: they highlight its consequences

for phenomena of interspecific population dynamics such as synchrony in breeding performance.

L256 I would also add somewhere toward the end a caveat regarding the interplay between trends in (breeding) population size, breeding performance, and (anthropogenic and non-anthropogenic) resource availability, particularly in open populations where it is unclear whether fluctuations in resource availability translate more into shifts in species' range or population size.

Figure 1 the letters indicating the subsections of the figure (a;b;c) are not represented on the figure. The bird drawings in the legend are not very informative, especially for the reader unfamiliar with how these species look like, perhaps the species name could be added here?

Figure 2 please state what the line and greyed area represent

Table 1 I don't see “np” in the table, but the "df" field is not referred to in the caption. same comment applies to table C.1

Table A.1 So these are the same values represented in figure 1? Can you please provide the SE for egg volumes?

Table C.2 For Sync and NAO, would it be possible to report on estimated marginal means instead of estimated differences from the baseline values?

See https://cran.r-project.org/web/packages/emmeans/vignettes/models.html#I for compatibility with the used modelling procedure

I’d finally like to thank the authors for sharing the code for their analyses.

Reviewer #2: This study aims to look at the synchrony of breeding investment in two seabird species following a change in food subsidies. While the question is relevant and interesting to the field, the authors fail to clearly presents how the address the question and how they answer their hypothesis. I also have major concerns on the statistical analysis. I appreciated the supplementary material which were very informative, but I think some of this information has their place in the manuscript itself. I suggest the manuscripts needs major revisions prior to publication.

Major comments:

1. Authors brings this idea of subsidy-decoupling hypothesis, yet fail to address it in their own study hypothesis, and prediction but most especially in the discussion. Most importantly, they fail to clearly show how their analysis answers the hypothesis or research questions. Authors use a lot of scientific terms without defining them or giving concrete examples that could help an uninformed, novice and even experienced readers to understand the claims made in the study.

2. Authors use mean egg volume per year as their experimental unit, as a proxy to breeding investment. I think this is concerning in several ways:

a. Gulls lay three eggs. Therefore, those three eggs are not independent from each other, as they are part of the same breeding investment from the same gull pair. Not only this is a problem on itself, but the size of the egg may very with the order they are laid, most especially the third egg which is often smaller than the first two. This may lower the mean egg volume per year.

b. Shearwaters only lay one egg. Comparing the volume of that one egg with eggs produced within a clutch is not statistically correct as sampling units differ between species. You could use the volume of the largest egg per gull clutch or calculate a volume per nest (sum of all egg volumes?). These would be a comparable measures of breeding investment with the single shearwater egg.

c. By measuring only three egg clutches, you are not really quantifying the breeding investment of your population, just the ones of the individuals who invests the most. From year to year, the proportion of three versus two egg clutches may vary. To fully compare the breeding investment of gulls from year to year, variation in clutch size would also be indicative of breeding investment in your population.

d. While your best models (the linear model with WNAO) do not include year in the model, including year and species in the same model as seen in Table C.1 (where mean egg volume per year is your response variable) is pseudoreplication (1 measure of egg volume per year and per species) and therefore is not valid. I’m also wondering why you calculated a mean per year (thus using year as the experimental unit) instead of using the nest as your experimental unit.

Minor comments:

Abstract

Consider rewriting your abstract with statistical results and your hypothesis (and whether you validated them).

Introduction

Line 46-52: Not relevant information, especially to start your introduction

Line 53-67: You talked about what affect synchrony here, yet I still don’t know what you mean by synchrony (definition at line 106-107) or what studying synchrony really brings to understanding of population and interspecific dynamics. These answers come later in the intro, I suggest putting them earlier

Line 64: … density of individuals fluctuates indenpendantly … of what? Environment?

Line 75: Add examples of demographic parameters as opposed to just breeding parameters later in the paragraph

Line 81-82: Anthropogenic food subsidies shape communities … How?

Line 87-81: Sentence hard to read. Does give us concrete example on how the patterns are influenced or the expected effects on survival and reproductive parameters.

Line 94: You are referring here to Figure 1, which is part of your results, before even stating an hypothesis. Remove mention of figure 1.

Line 110 : suggest: two sympatric avian top predators

Line 113: Why shortening Yellow-legged gull to Lm rather than just gull and Scopoli’s shearwater to Cd instead of just shearwater? Seems like if we miss that part of the intro, the rest of the paper is confusing.

Line 137: increasing the degree of synchrony in egg volume relative to years with an active landfill. I would also had a prediction about environmental buffering, which you test with general linear model, but don’t mention here with your hypothesis.

Methods

Line 151-153: Using egg volume as a proxy to breeding investment should be clear in your hypothesis as well.

Line 155-164: I’d suggest beefing up this part a little bit with some of the information provided in the supplementary material. You explain what cross-correlation and State-based Markov Chain modelling work and are relevant in your study, but not really how you used them with your data or what measure you get out of it.

Line 167-170: This part should be in the introduction

Line 175: Where is the random effect of your mixed model? According to your supplementary material, it looks like a linear model without random effects.

Line 180-181: If two model had an ΔAIC < 4, which one did you choose ultimately as your best model? The most parsimonious? By looking at table C2, looks like you averaged the estimates of your candidate model (yet no mention of this in the methods). Yet again, Table 1 shows ‘ the best model’ in bold. What are your criteria, just the AIC?

Result

Line 185-186: Can you statistically test the difference between your CC? Is there an error associated with this measure?

Line 189: AIC or AICc? Methods doesn’t mention that you corrected for small sample size.

Lines 190-192: Indicate that the values in bracket are the confidence interval and that you considered a significant effect when CI did not cross 0 in your methods.

Discussion

Line 205: This is the first (and only) mention of entropy analysis in the manuscript (excluding supplementary material). Make sure to introduce the term earlier.

Line 208-210 : This is the only sentence discussing the environmental buffering tested with your linear model. Why is it due to low sample size? No other reasons? Maybe it is because you included shearwaters (which were not affected by landfill closure) instead of testing gulls by themselves?

Line 215: Reference?

Line 218: evidence for what?

Line 241: What in your result specifically indicates transient dynamics?

Line 245: What sis missing in your study to pin-down the factors responsible to synchronization? What would you suggest for further studies?

References

Lines 314-318: This paper seem duplicated.

Figure 1:

The different states are not described in the methods (only in the supplementary material). It needs to be described for better interpretation.

Why 1.96 SE instead of 95% CI ?

Figure 2:

This figure is misleading of results since WNOA was only significant as an additive effect, thus the curve should be all points, no separation by species or synchrony.

If kept as is, put a) and b) in the same panel and c) and d) in the same panel with different shades and point shapes. Put error of data points (since it’s the annual mean in egg volume).

Table 1 and Table C.1:

-Why is synchrony in the model and not landfill closure? I know you proved that synchrony is tied to landfill closure, but here, you want to test environmental buffering before and after landfill closure and it seems to me, this variable would be more representative (and informative) than just synchrony.

-Why include shearwaters if they were not affected landfill closure? Your question is whether environmental buffering was present during landfill closure (so the difference of the Vegg~WNAO curve before versus after landfill closure). This could be answered with a simple Vegg ~ WNAO*landfill closure using the gull data only.

-Seems like you just tested all combinations of variable without consideration for biological process and relevance. See my previous comment on pseudoreplication.

Table C.2:

Indicates what bold font means

6. PLOS authors have the option to publish the peer review history of their article (what does this mean?). If published, this will include your full peer review and any attached files.

Reviewer #1: **Yes: **Alejandro Sotillo

Reviewer #2: No

---

## [Author Response · Author response to Decision Letter 0]

3 Aug 2022

We thank the reviewers for their thorough and encouraging comments. We have included detailed responses below and as an attached file . Notice some of the responses include figures that are not supported in the web form so we would kindly refer the reviewers to the attached document for full answers. We think our manuscript has greatly improved thanks to your suggestions and comments and we look forward to hearing from you. 

Reviewers' comments:

Reviewer's Responses to Questions

Comments to the Author

1. Is the manuscript technically sound, and do the data support the conclusions?

Reviewer #1: Yes

Reviewer #2: Yes

2. Has the statistical analysis been performed appropriately and rigorously?

Reviewer #1: Yes

Reviewer #2: No

3. Have the authors made all data underlying the findings in their manuscript fully available?

Reviewer #1: Yes

Reviewer #2: Yes

4. Is the manuscript presented in an intelligible fashion and written in standard English?

Reviewer #1: Yes

Reviewer #2: Yes

5. Review Comments to the Author

Please use the space provided to explain your answers to the questions above. You may also include additional comments for the author, including concerns about dual publication, research ethics, or publication ethics. (Please upload your review as an attachment if it exceeds 20,000 characters).

Reviewer #1: I’d like to thank the authors for creating this interesting article, that takes an original look at the phenomenon of anthropogenic food subsidies in vertebrate populations, adding the community aspect, which is rarely assessed in this context. I enjoyed reading it, it is well-written, and I believe the methods are sound. My main criticism would be that the time series is still too short to confirm certain trends that are suggested in the discussion. On the other hand, we can’t wait for decades to report on demographic monitoring trends when they concern aspects relating to the short and mid-term consequences of policy changes on wildlife. I refer to my detailed comments on the discussion for what I think might be a bit weak in terms of conclusions. I believe this manuscript is fit for publication after minor revision. I look forward to a revised version. 

#Authors: We appreciate the reviewers’ encouraging comments and their suggestions for improving our manuscript. 

L76 ‘considerably’

#Authors: replaced “considerable” with “considerably’”

Detailed comments:

L46-67 This general introduction to synchrony, and its focus on intra-specific, inter-population synchrony, is a bit too long and far-reaching. Could the authors try to shorten it, perhaps into a single paragraph?

#Authors: we have now shortened the introduction following both reviewers' 1 and 2 suggestions. 

L68 replace “have” by “having” or “despite that”

#Authors: replaced “have” with “having”

L76 ‘considerably’

#Authors: replaced “considerable” by “considerably’”

L78-79 I don't see how one follows the other: synchrony is a useful indicator of environment state because breeding parameters reflect environmental conditions?

#Authors: Previous studies, e.g. Lahoz et al. 2013, argue that multispecies synchrony of productivity can be used as community-based integrative indicators of the marine ecosystem. This is because it integrates information from different species and it might allow detecting changes in the ecosystem without having to measure environmental conditions for each one of the species considered.

L84 “These subsidies”

#Authors: replaced “There” by “These”

L89; 95-96 I suggest to avoid an ambiguous use of the words “nature” and “natural”

#Authors: We agree and removed “natural in line 89 and replaced it by “wild” in 95. 

L92 perhaps replace “species” by “populations”?

#Authors: We agree and replaced “Species” with “populations”

L 97 The term "subsidy" has other connotations when used in combination with "EU" (i.e. economic incentives). How about replacing it by "waste" or "food waste"?

#Authors: We agree. We have now replaced “food subsidy” with “food waste”

L101 “aiming for the termination”

#Authors: Thanks. We have now replaced “aiming termination” with “aiming for the termination”

L103 “a ban on fisheries discards”

#Authors: We replaced “fisheries discard ban” with “a ban on fisheries discards”

L104 delete “experimentally”. Also “aim to modify”

#Authors: deleted “experimentally” and replaced “aim modify” with “aim to modify”

L105 “of anthropogenic food subsidies for wildlife” Please be generous with the non-specialized reader

#Authors: added “of anthropogenic food subsidies for wildlife”

L111 “waste becoming inaccessible”

#Authors: added “becoming”

 L124 sympatry: if the nesting grounds do not overlap, do the home ranges overlap? How much does the spatial niche of these species actually overlap?

#Authors: This is an interesting point. We have thorough information on the home range of Scopoli’s shearwaters as several studies have been collecting GLS data from them. However, the collection of gull movement data in the study colony began recently. We know that both species feed behind fishing vessels during the discard of unwanted catch. We also know that home-ranges can overlap along coastal areas, however the extent of home-range overlap is not known. This is partly why we can’t disentangle the ultimate mechanism driving the synchrony but we agree it is an interesting question.

L152 do we have any evidence for a relationship between maternal investment and breeding success in these species? If not, references to works on similar species may instead be cited

#Authors: We do not have experimental evidence on the relationship between maternal investment and breeding success. However, in gulls, the egg size varies with food availability (see for example Steigerwald et al. 2015 cited in the text) and in Cory’s Shearwaters, egg-size correlates with breeding success (see the figure below from our unpublished data as an example). 

Whether birds are restrained or constrained cannot be answered without experimental data. In other species of birds, experimental data have demonstrated that breeding success and investment (i.e. egg size) relate to parental quality and egg volume (e.g. Bolton 1991). We have now mentioned these studies and added the respective references.

L158 replace ‘despite’ by ‘although’

#Authors: replaced ‘despite’ by ‘although’

L159 replace “necessary” by “necessarily”

#Authors: replaced “necessary” by “necessarily”

L172 delete “we used we used the”

#Authors: deleted “we used we used the”

L181-183 do you mean by this that you included the interaction between NAO and your proxy for synchrony in the full model? Please state explicitly which interaction were included and to what end

#Authors: No. We coded a new variable from the annual synchrony value where the years with synchrony were coded as 1 and the others with a 0. We have now further explained this in the methods. 

L187; 206; 223 I’m afraid I’m misinterpreting the figure… but it seems like synchrony is observed only in 5 years after landfill closure? But you consistently mention a 7-year synchrony period throughout the manuscript. Can you please clarify this to me in the rebuttal letter?

#Authors: The way the state-based Markov chain modelling works is that for each time-series we built a new one {Yt} of k-2 elements (with t = 2, 3, ..., k-1). The series {Yt} describes the variable state at a given point in time in relation to the precedent and subsequent states. What we are visualizing in the figure is the Yt time-series where you see 5 points actually represent 7 years. We have provided further explanation to make this clearer in the figure text.

L209 “decreased”

#Authors: replaced “decrease” by “decreased”

L210 replace “were” by “was”

#Authors: replaced “were” by “was”

L211 Perhaps be a bit more explicit on what this event was: "After closure of the open-air landfill, and consequent disappearance of a main local source of anthropogenic food subsidies, ..."

#Authors: We agree and we have now replaced “After subsidies cessation” with “After closure of the open-air landfill, and consequent disappearance of a main local source of anthropogenic food subsidies,”

L213 – 214 ‘the foraging strategy of one species’: I’d rather explicitly say here “of the species so far relying on the disappeared subsidies”.

#Authors: We have replaced “of one species” with “of the species so far relying on the disappeared subsidies”. 

I feel like the more interesting trends are here given by Lm, while the discussion is treating both species equally.

#Authors: We agree. The closure of the landfill was a major perturbation for gulls but not for shearwaters. The effect of landfill closure on LM ecology has been described in two main works (Steigerwald et al. 2015 and Payo Payo et al. 2017; both cited in the text). Here we wanted to investigate whether the perturbation has driven synchrony that the availability of anthropogenic waste had decoupled. In this respect, CD is equally important because it sets the dynamic of an “undisturbed” process.

L214 “and/or”

#Authors: replaced “or/and” by “and/or”

L214-215 do the numbers (nLm and nCd) reported in Table A.1 reflect total population size of the study colonies? Are there discernible trends in these population sizes, particularly in relation to the closure of the landfill?

#AUTHORS: No. nLm and nCd are the number of eggs measured by year and species. They do not reflect the population size. We have population size proxy estimates for the Yellow-legged gull. They decreased after the landfill closure in 2010 but progressively increase since 2012, reaching a stable (lower) number in 2014-15. The egg size in gulls at Dragonera had a similar trend and the two variables are positively correlated. 

A positive density-dependent relationship between egg size and nest number suggests that animals are constrained by food availability and that egg volume is related to resource availability. Scopoli’s shearwater population size is difficult to estimate. The number of nests monitored remained constant through the study period and it is not a good proxy of population size. 

L223 ‘time associated with acquiring’

#Authors: included “with”

L224 so this reported synchrony is interpreted as symptomatic of a decreased efficiency in the use of the surrounding landscape by the Lm's nesting in Dragonera?

#Authors: We interpret it as a ‘learning period’ during which the individuals that used human waste begin to search and use alternative resources. We also believe that part of the gulls that foraged at the landfill had moved to other sites on the Iberian Peninsula. The gulls that remained were those that already fed on fishery discards or have been able to learn how to do it. The number and quality of gulls that are now breeding at Dragonera Island after the landfill closure might well be the result of both processes, individual improvement and selection.

L231 so marine productivity increases under more positive NAO conditions? It would be helpful if either in the introduction or earlier in the discussion a short explanation is provided on what NAO is and what it most probably implies from the point of view of breeding seabirds.

#Authors: We agree. We have now included in the main text the explanation of the relationship between WNAO and food availability for seabirds. 

L233 “carry-over”

#Authors: replaced “carrying over” with “carry-over” 

L235 What would be the mechanism linking population size change to synchrony in maternal investment? I fail to understand how the latter would be a consequence of the former without more details. Is it that Lm reproductive individuals exploiting garbage migrate away from the population, with only those Lm individuals that respond in their reproductive performance similarly to Cd remaining in the colony? I feel like abstract terminology is sometimes getting in the way of the interpretation of results in this discussion.

This is a complicated issue. We admit that we do not have a simple answer. The selection hypothesis, suggested by the referee, is a likely explanation (see the answer to the comment above). It agrees with the observed decrease in the population size of LM. However, other, more complicated mechanisms might be involved. For example, in gulls, males feed females during the courtship. This might modulate the maternal investment concerning food abundance and mate quality. The positive correlations between egg average volume and population density in LM and between volume and breeding success in CD (see figures above) reveal a complicated link between food abundance, maternal investment, population size and breeding success. A link that we do not fully understand at the moment.

L239 is there any reason to suspect a similar (interspecific interaction) mechanism to be at play in this context? If so, which one exactly? Feeding competition at sea?

#Authors: Yes. As we have stressed in the text we were not able to pin down a mechanism but the most likely one is indeed interspecific competition. We have now included the sentence “ and changes in interspecific competition at sea could be underlying this pattern”

L249 "non-linear dynamics": Please be more explicit on exactly what you claim to prove. Are you saying that we have evidence that the sudden disappearance of food subsidies induces in itself a response in population dynamics that would not be expressed if this disappearance had been gradual? What is this claim based on?

#Authors: We agree that this term can lead to confusion. We have now removed “non-linear”. 

L252 Did we previously not think that anthropogenic food subsidies buffer natural variability in population dynamics of the species exploiting them? And that this had consequences on the community dynamics? 

#Authors: Yes

In the introduction it was expected, based on previous knowledge, that the population so far relying on subsidies would respond to environmental fluctuations more similarly to a population that did not rely on subsidies, because these subsidies decouple population dynamics from non-anthropogenic environmental variation. 

#Authors: Correct

I would rather argue that the results confirm this already existing theory from a novel point of view: they highlight its consequences for phenomena of interspecific population dynamics such as synchrony in breeding performance.

#Authors: We agree. This is indeed the novelty of our work. Beside buffering natural selection and have a direct influence on population dynamics, anthropogenic subsidies decouple synchrony. In this case we measured egg-volume because we do not have access to the total number of Shearwater breeding pairs and took it as a proxy of breeding investment and food intake/availability.

L256 I would also add somewhere toward the end a caveat regarding the interplay between trends in (breeding) population size, breeding performance, and (anthropogenic and non-anthropogenic) resource availability, particularly in open populations where it is unclear whether fluctuations in resource availability translate more into shifts in species' range or population size.

#Authors: In the case of LM we know that the closure of the landfill led to a significant decrease in breeding parameters, population size and individual body condition (Steigerwald 2015 and Payo-Payo et al 2016).

Figure 1 the letters indicating the subsections of the figure (a;b;c) are not represented on the figure. The bird drawings in the legend are not very informative, especially for the reader unfamiliar with how these species look like, perhaps the species name could be added here?

#Authors: We have now modified the figure to include panel labels and the names of the species. 

Figure 2 please state what the line and greyed area represent

#Authors: We have now included “model predictions (black line with 95% confidence grey bands)”

Table 1 I don't see “np” in the table, but the "df" field is not referred to in the caption. 

#Authors: Thank you for spotting this. We have now included definition of df and removed definition of np.

same comment applies to table C.1

#Authors: We have now included definition of df and removed definition of np.

Table A.1 So these are the same values represented in figure 1? Can you please provide the SE for egg volumes?

#Authors: We have now updated the table to include the SE as requested.

 Table C.2 For Sync and NAO, would it be possible to report on estimated marginal means instead of estimated differences from the baseline values? See https://cran.r-project.org/web/packages/emmeans/vignettes/models.html#I for compatibility with the used modelling procedure. I would finally like to thank the authors for sharing the code for their analyses.

#Authors: We were not completely sure about what the reviewer is asking us to do. We are providing what we think they meant but we are keen to provide more information if this is not what they were looking for. In essence, we present aliased parameters resulting from the averaging of the top competing models. However, we would still prefer to maintain the original table in the main manuscript.

Parameter Estimate 95% CI

Cd 73.09 [72.20, 73.99]

Lm 73.66 [73,6575.44]

SyncYES 4.483 [-33.02, 41.99]

WNAO -0.388 [-0.69, -0.08]

Lm:YES -2.014 [-3.85, -0.18]

Lm:WNAO 0.191 [-0.26, 0.64]

SyncYES: WNAO -0.015 [-0.58, 0.55]

SyncNO 73.825 [73.17,74.47]

We apologize if this is not what the reviewer was asking for and we would be happy to address any further comments or if they would such as further post hoc with standardized variables, pairwise comparisions and contrasting effect. Overall, we thank the reviewer for their constructive and thorough comments throughout our manuscript and their encouraging reviewing style. It is much appreciated. 

Reviewer #2: This study aims to look at the synchrony of breeding investment in two seabird species following a change in food subsidies. While the question is relevant and interesting to the field, the authors fail to clearly presents how the address the question and how they answer their hypothesis. I also have major concerns on the statistical analysis. I appreciated the supplementary material which were very informative, but I think some of this information has their place in the manuscript itself. I suggest the manuscripts needs major revisions prior to publication.

Major comments:

1. Authors brings this idea of subsidy-decoupling hypothesis yet fail to address it in their own study hypothesis, and prediction but most especially in the discussion. Most importantly, they fail to clearly show how their analysis answers the hypothesis or research questions.

#Authors: We agree. We have now expanded the hypothesis formulation and predictions at the end of the introduction. We have also changed part of the discussion to focus on the starting hypotheses.

Authors use a lot of scientific terms without defining them or giving concrete examples that could help an uninformed, novice and even experienced readers to understand the claims made in the study.

#Authors: We have now revised the manuscript looking for such terms and included some extra definitions. However, without relevant line numbers and examples of the scientific terms the reviewer is referring too, it is difficult to address this comment. 

2. Authors use mean egg volume per year as their experimental unit, as a proxy to breeding investment. I think this is concerning in several ways:

a. Gulls lay three eggs. Therefore, those three eggs are not independent from each other, as they are part of the same breeding investment from the same gull pair. Not only this is a problem on itself, but the size of the egg may very with the order they are laid, most especially the third egg which is often smaller than the first two. This may lower the mean egg volume per year.

b. Shearwaters only lay one egg. Comparing the volume of that one egg with eggs produced within a clutch is not statistically correct as sampling units differ between species. You could use the volume of the largest egg per gull clutch or calculate a volume per nest (sum of all egg volumes?). These would be a comparable measures of breeding investment with the single shearwater egg

#Authors: We agree with referee’s point. Eggs from the same clutch are not independent from one another. However, average, maximum and total egg-volume within 3-egg clutches are highly correlate. In fact, Steigerwald et al (2015; cited in the text) already demonstrated in the same species and breeding colony that results are consistent when using one or the other (supplementary information in Steigerwald et al. 2015). 

To further investigate this, we provide below the relationship between max, average and total egg volume in 3-egg clutches (n=425), measured between 2007-2022 at Dragonera Island. 

The R-square measured of the correlations ranged between 0.829 and 0.999. The absolute value of the average egg is indeed lower than the largest egg in the clutch, however, given the high correlation, it will not change the yearly comparison used in our analysis. 

c. By measuring only three egg clutches, you are not really quantifying the breeding investment of your population, just the ones of the individuals who invests the most. From year to year, the proportion of three versus two egg clutches may vary. To fully compare the breeding investment of gulls from year to year, variation in clutch size would also be indicative of breeding investment in your population.

#Authors: Correct. We are aware that our measure of breeding investment is not perfect. However, by using 3-egg clutches we use the best possible proxy. In fact, 2-egg clutches might not be completed or might have been reduced by predation. Using 3-egg clutches, we reduced the uncertainty regarding the timing of the survey and therefore minimize the use of incomplete clutches or predated nests. 

d. While your best models (the linear model with WNAO) do not include year in the model, including year and species in the same model as seen in Table C.1 (where mean egg volume per year is your response variable) is pseudoreplication (1 measure of egg volume per year and per species) and therefore is not valid. I’m also wondering why you calculated a mean per year (thus using year as the experimental unit) instead of using the nest as your experimental unit.

#Authors: We agree. We did include the annual mean of the mean egg volumes by clutch but we hadn’t properly explained it in the methods. We have now added this information to the methods. 

Minor comments:

Abstract

Consider rewriting your abstract with statistical results and your hypothesis (and whether you validated them).

#Authors: We have now modified the abstract to include the suggestions of the reviewer. 

Introduction

Line 46-52: Not relevant information, especially to start your introduction

#Authors: We have modified the introduction following the suggestions of both reviewers. 

Line 53-67: You talked about what affect synchrony here, yet I still don’t know what you mean by synchrony (definition at line 106-107) or what studying synchrony really brings to understanding of population and interspecific dynamics. These answers come later in the intro, I suggest putting them earlier

#Authors: We agree. We have modified the introduction and moved the synchrony definition earlier in the manuscript. 

Line 64: … density of individuals fluctuates indenpendantly … of what? Environment?

#Authors: We agree this phase could have led to confusion. We have now modified the text in the first two paragraphs to accommodate previous requests from both reviewers. 

Line 75: Add examples of demographic parameters as opposed to just breeding parameters later in the paragraph

#Authors: The references include Lahoz et al. 2011, which focuses on other demographic parameters such as survival

Line 81-82: Anthropogenic food subsidies shape communities … How?

#Authors: We included “by altering processes such as competition, predator–prey interactions”. More explanation can be found in Oro et al. 2013 (cited in the text).

Line 87-81: Sentence hard to read. Does give us concrete example on how the patterns are influenced or the expected effects on survival and reproductive parameters.

#Authors: We agree. We have now rephrased the sentence to help the reader. 

Line 94: You are referring here to Figure 1, which is part of your results, before even stating an hypothesis. Remove mention of figure 1.

#Authors: We agree. Thank you for signalling this. We have removed the mention of figure 1. 

Line 110 : suggest: two sympatric avian top predators

#Authors: We have included “avian” 

Line 113: Why shortening Yellow-legged gull to Lm rather than just gull and Scopoli’s shearwater to Cd instead of just shearwater? Seems like if we miss that part of the intro, the rest of the paper is confusing.

#Authors: We have replaced “Lm” by “Yellow-legged gull” and “Cd” by “Scopoli’s Shearwater”.

Line 137: increasing the degree of synchrony in egg volume relative to years with an active landfill. I would also had a prediction about environmental buffering, which you test with general linear model, but don’t mention here with your hypothesis.

#Authors: We have now included “Moreover, we expect food subsidies should buffer egg volume — as a proxy of breeding investment — against environmental variation.”

Methods

Line 151-153: Using egg volume as a proxy to breeding investment should be clear in your hypothesis as well.

#Authors: We agree. We have now included it in our hypothesis. See previous comment answer. 

Line 155-164: I’d suggest beefing up this part a little bit with some of the information provided in the supplementary material. You explain what cross-correlation and State-based Markov Chain modelling work and are relevant in your study, but not really how you used them with your data or what measure you get out of it.

#Authors: We have now moved the explanation of the State-based Markov chain modelling into the main text to help the reader navigate through the Methods section. 

Line 167-170: This part should be in the introduction

#Authors: We have now included this at the end of the introduction. 

Line 175: Where is the random effect of your mixed model? According to your supplementary material, it looks like a linear model without random effects.

#Authors: That is correct we have now modified the methods accordingly. 

Line 180-181: If two model had an ΔAIC < 4, which one did you choose ultimately as your best model? The most parsimonious? By looking at table C2, looks like you averaged the estimates of your candidate model (yet no mention of this in the methods). Yet again, Table 1 shows ‘ the best model’ in bold. What are your criteria, just the AIC?

#Authors: Good point. We have now included the explanation of how we use the AIC to select models to be included in the model averaging. The models shown in table 1 are those within 4 points but we only highlight with bold font the one with the lowest AIC. 

Result

Line 185-186: Can you statistically test the difference between your CC? Is there an error associated with this measure?

#Authors: We calculated the z-scores of the CC and performed a test for the difference of the two samples. The difference has a support for the difference with a p-value of 0.1. We would prefer not to include this in the main text as we think it would be more correct to test this assumption within the GLM framework. However, we have preferred to include synchrony rather than before-after as an explanatory variable. 

Line 189: AIC or AICc? Methods doesn’t mention that you corrected for small sample size.

#Authors: Good point. We have now included “corrected for small sample size” in the methods section. 

Lines 190-192: Indicate that the values in bracket are the confidence interval and that you considered a significant effect when CI did not cross 0 in your methods.

#Authors: We have now included confidence intervals in the results section and specified that “we consider an effect significant when the CI of the beta did not overlap with zero”.

Discussion

Line 205: This is the first (and only) mention of entropy analysis in the manuscript (excluding supplementary material). Make sure to introduce the term earlier.

#Authors: We agree. We have now referred to entropy in the methods section. 

Line 208-210 : This is the only sentence discussing the environmental buffering tested with your linear model. Why is it due to low sample size? No other reasons? Maybe it is because you included shearwaters (which were not affected by landfill closure) instead of testing gulls by themselves?

#Authors: Our model includes the interaction with the species therefore we would be able to differentiate for different patterns in the two species if it was present and we had enough sample size. However, we are unable to detect significantly detect this with the current data. 

Line 215: Reference?

#Authors: We have now included the citation Payo-Payo et al., 2015.

Line 218: evidence for what?

#Authors: We have now reordered the sentence. “Although confirming evidence for increased diet overlapping of the two species is lacking, diet shift could potentially have resulted in increasing interference competition”

Line 241: What in your result specifically indicates transient dynamics?

#Authors: we have now included “as synchrony disappears after 7-years”

Line 245: What sis missing in your study to pin-down the factors responsible to synchronization? What would you suggest for further studies?

#Authors: Good point. The weakness of our study certainly is that we are not able to pin-down a given mechanism unequivocally. Nonetheless, we bring evidence to demonstrate that the closure of the landfill generates synchronous variations in the egg volume of these two seabirds. We think that future work on direct tracking and isotopes analyses can shed a light on diet and home-range overlap between the two species to measure the level of intra-specific competition.

References

Lines 314-318: This paper seem duplicated.

#Authors: Well spotted. We have removed the second one. 

Figure 1:

The different states are not described in the methods (only in the supplementary material). It needs to be described for better interpretation.

#Authors: We have now included the description of the states in the methods. We hope this will help the reader navigate the figure. 

Why 1.96 SE instead of 95% CI ?

#Authors: We have now replaced “1.96 SE” by “95% CI”

Figure 

This figure is misleading of results since WNOA was only significant as an additive effect, thus the curve should be all points, no separation by species or synchrony.

If kept as is, put a) and b) in the same panel and c) and d) in the same panel with different shades and point shapes. Put error of data points (since it’s the annual mean in egg volume). 2:

#Authors: We have now edited the figure as suggested. 

Table 1 and Table C.1:

-Why is synchrony in the model and not landfill closure? I know you proved that synchrony is tied to landfill closure, but here, you want to test environmental buffering before and after landfill closure and it seems to me, this variable would be more representative (and informative) than just synchrony.

#Authors: We understand the rationale of the reviewer, however after the landfill was closed the synchrony only lasted for 7 years after which the synchronization disappeared. It is possible that gulls found some alternative food to minimize competition or simply to reverse a diet dominated by waste that can be found opportunistically in parks, harbours or near restaurants. However, we do not have a clear picture of the foraging strategy of the two species. After the landfill was closed there is a mix of years with and without synchrony that we think will mask the environmental buffering. That is why as both variables are very correlated we decided to use synchrony.

-Why include shearwaters if they were not affected landfill closure? Your question is whether environmental buffering was present during landfill closure (so the difference of the Vegg~WNAO curve before versus after landfill closure). This could be answered with a simple Vegg ~ WNAO*landfill closure using the gull data only.

#Authors: We agree. However, we were interested in whether the two time-series exhibit synchrony after landfill closure. By including the interaction term across species we should be able to detect that same pattern the reviewer is describing. 

-Seems like you just tested all combinations of variable without consideration for biological process and relevance. See my previous comment on pseudoreplication.

#Authors: Not sure to have fully understood this point. Pseudoreplications can only be present among eggs from the same nest. However, this is not a real problem because maximum volume, total volume and average volume are highly correlated (see figures above).

Table C.2: Indicates what bold font means

#Authors: This information has now been included. 

6. PLOS authors have the option to publish the peer review history of their article (what does this mean?). If published, this will include your full peer review and any attached files.

Do you want your identity to be public for this peer review? For information about this choice, including consent withdrawal, please see our Privacy Policy.

Reviewer #1: Yes: Alejandro Sotillo

Reviewer #2: No

---

## [Decision Letter · Decision Letter 1]

26 Aug 2022

PONE-D-22-11852R1Interspecific synchrony on breeding performance and the role of anthropogenic food subsidies.PLOS ONE

Dear Dr. Payo Payo,

Thank you for submitting your manuscript to PLOS ONE. After careful consideration, we feel that it has merit but does not fully meet PLOS ONE’s publication criteria as it currently stands. Therefore, we invite you to submit a revised version of the manuscript that addresses the points raised during the review process.

We look forward to receiving your revised manuscript.

Kind regards,

Jose M. Riascos, Ph.D.

Academic Editor

PLOS ONE

Journal Requirements:

Reviewers' comments:

Reviewer's Responses to Questions

**Comments to the Author**

1. If the authors have adequately addressed your comments raised in a previous round of review and you feel that this manuscript is now acceptable for publication, you may indicate that here to bypass the “Comments to the Author” section, enter your conflict of interest statement in the “Confidential to Editor” section, and submit your "Accept" recommendation.

Reviewer #1: (No Response)

Reviewer #2: All comments have been addressed

2. Is the manuscript technically sound, and do the data support the conclusions?

Reviewer #1: Partly

Reviewer #2: Yes

3. Has the statistical analysis been performed appropriately and rigorously? 

Reviewer #1: Yes

Reviewer #2: Yes

4. Have the authors made all data underlying the findings in their manuscript fully available?

Reviewer #1: Yes

Reviewer #2: Yes

5. Is the manuscript presented in an intelligible fashion and written in standard English?

Reviewer #1: Yes

Reviewer #2: Yes

6. Review Comments to the Author

Reviewer #1: Main comments

I thank the authors for the care and seriousness they have shown in addressing every single comment that the reviewers made in the first round. I find that the manuscript has improved and the authors have addressed my concerns rather satisfactorily. The study appears now more transparent, and I get a better grasp of the hypotheses, results and conclusions. The methods section has greatly improved, but it is still missing some important information.

In relation with the comments by reviewer #2 about the quality of the analyses performed, it would illustrate the representativeness of the sample that was analyzed if yearly estimated nesting population size, number of sampled nests and proportion of these nests that contained 3 eggs were presented. Is there any tendency between years in clutch size? Were nests monitored once per year or several times? When exactly were they monitored? Is there any information on interannual variation in reproductive phenology (advanced/delayed – concentrated/spread out laying seasons)? If concepts such as synchrony, sympatry and interspecific competition are invoked, it should be clearly acknowledged that the temporal niche does not fully overlap (interspecific differences in laying phenology). What is the difference in median laying date between the 2 species? Did it vary between years?

I have answered "partly" to the question "2. Is the manuscript technically sound, and do the data support the conclusions?" because, regarding the message on environmental buffering by anthropogenic food subsidies, the results as they are now presented do not convey this idea just by themselves. I trust the authors’ claim, but it is just not very striking from the figures and tables that they show. I suggest the authors reconsider how they want to present their evidence for this important point they make in their paper. If evidence is weak, they might want to consider removing the focus from it.

Comments on the authors’ rebuttal

Reviewer 1 “L124 sympatry: the extent of home range overlap is not known” This is not necessarily a problem, but it would be good to state this clearly: that sympatry of these populations is assumed rather than well-documented.

Reviewer 1 “L214-215” Thank you for this information. Could this be made available in the supplementary material?

Reviewer 2 “2. Authors use mean egg volume per year as their experimental unit, as a proxy to breeding investment. I think this is concerning in several ways:” Thank you for this information. Could this be made available in the supplementary material? Also, if you have information on interannual variation of clutch sizes across the study period, please provide this in the sup mat. I understand that some of this information may be found in other publications based on the same data base, but such details in annex do help the present paper stand on its own.

Comments and suggestions on the revised manuscript:

L23 I'm not sure that metapopulations are relevant enough in this case to be mentioned in the abstract, I suggest removing “metapopulations persistence”.

L35 “environmental buffering…” I'm not convinced that sufficient evidence is provided to support this claim. If the authors believe there is, they should try to highlight it more in the results (see main comments as well as comments below).

L50 replace “the correlation” by “correlating”

L63 remove “- such as egg volume –“

L64 replace “so the” by “, making”. Remove “could be used as”

L66 remove “new”

L68 “competition and predator-prey interactions”

L69 “worldwide, and”

L71 remove “potentially”

L72 “and/or”

L78 “should” instead of “would”

L80;81 does it reveal dynamics that were otherwise hidden, or does it restore dynamics that had been altered through anthropogenic impact?

L85 “decades”

L86 I'm not sure that the subsidization of some wildlife populations had a very relevant part in motivating any of both directives. In my understanding, the landfill directive was responding to a concern about environmental pollution, while the discards ban was about tackling overfishing to retain only larger-sized fish, lack of selectivity in fishing methods and a pointless source of waste. I'd rather say that the reduction of this subsidization phenomenon was a positive side-effect. But please ignore this point if the cited legislation actually states it as a main motivation.

L90 replace “is a fantastic” by “provides an”

L91 replace “ultimately aim to” by “will”

L94 replace “changes” by “variability”

L97 remove the commas around “regularly”

L99 remove the comma before anthropogenic

L107 do you mean adult survival probability?

L108 and elsewhere: I'd be rather inclined to write "The Yellow-legged Gull", particularly when opening a sentence

L113 replace “have” by “having”

L117 remove “- until 2010 –“

L145 remove “traditionally used and”

L189-191 The interactions that were considered in models should be mentioned here. The authors need to justify the biological meaning of adding these interactions

L201-202 I gather that the possible min-max range of CC is -1 to 1. Stating this explicitly gives an idea on how important this increase is.

L214 I fail to see what figure 2 is telling about environmental buffering. The authors should be more explicit as to what they are interpreting to be "environmental buffering". Is it the beta estimates for an interaction? Which one exactly? Where can the reader find these figures (which table)?

Discussion general comment: Now that WNAO has been described in methods as "High positive WNAO values are associated with the intensification of upwelling and small pelagic fish availability thus higher food availability for seabirds." I am missing a broader discussion on how the proxy for reproductive output in both species appears to be negatively correlated to this index.

L231 Sorry I don’t get the ending of the sentence: what similar drivers? Similar to what?

L234 No need to refer to figures in the discussion

L260 “is that landfill”

L272 “longer time”

L273 replace “clarification to” by “light on”

L276 see comment on L80;81

L277;278 I don't think you add any extra evidence on the pervasiveness of the impact of anthropogenic food subsidies. I'd rather say that the impacts are shown to express themselves in more compound ecological metrics such as inter-specific synchrony in breeding parameters

L280 replace “such as” by “like the” to avoid redundancy

L281;282 “or from sudden population collapses resulting from current environmental change” I find this very vague, can you be a little more specific? Do you mean that opportunities to study fluctuations in synchrony should be sought in circumstances where accelerated environmental change is known to be taking place? Like in areas suffering desertification, receding permafrost, deforestation…?

Supplementary material main comment: could the code be provided separately as a ".R" file? This way any user could just load your file instead of copy/pasting the code

Figure 2 Is it possible to add an in-figure color legend?

I thank the authors for their excellent work, and trust that after this second revision round any relevant issues will have been fixed. I’m looking forward to a second revised version.

Reviewer #2: I would like to congratulate the authors for their hard work on improving this manuscript and addressing the reviewer’s comments. Most of my concerns were address by simply clarifying the method which was greatly appreciated. The paper has improved significantly, and I only have minor suggestions.

L44: add ‘a’ before pronounced.

L50-51: Sentence is unclear. Suggestion: Intra-specific synchrony is apparent a at large spatial scale from correlated environmental conditions, the so-called ‘Moran’s effect’. I’m not sure if this is what you were trying to say.

L63-65: I’m having trouble connecting the start and end of this sentence. I suggest making it into 2 sentences: Breeding parameter – such as egg volume- often reflect conditions in the ecosystems. Therefore, using inter-specific synchrony of breeding parameters could be a useful indicator of local environment state.

Line 67: Your discussion now clearly answers your hypothesis, but I would mention your subsidy-decoupling hypothesis somewhere in your discussion.

Line 83-84: I don’t think a subsection is necessary here.

Line 93-123: This paragraph is very long. I suggest merging line 93-96 to previous paragraph and splitting the paragraph by species for the remaining lines (splitting somewhere around 108).

Line 108: Does YLG always breed close by to shearwaters or is that inly in the Mallorca region?

Line 121-123: I appreciate the addition; however, I’m still craving for more detail. Suggestion: add ‘so that egg volume is higher when environmental stochasticity is removed’ at the end of the sentence.

Line 137-138: Thank you for clarifying that you calculated the mean of egg volume per clutch prior to calculating the annual mean. Maybe mean egg volume per clutch instead? Was the annual mean used for both the synchrony and the GLM? At line 190-191, you only refer to ‘mean egg volume’ which suggest to me that you used mean egg volume per clutch and not the annual mean (which is appropriate for the GLM). If so, the annual mean should only be mentioned in the synchrony part. If not correct line 190-191.

Line 204: Add a sentence precising that synchrony stopped after 2017. This is a big part of the discussion but was not apparent to me until then.

Line 222: Correlation? Why not the actual relationship with you B estimate? Do you have an actual value of R2 to add to the figure?

Line 228-252: Long paragraph and a little confusing. I would first split at line 236. Then I would reshuffle the sentences around: all sentence mentioning overlap first, then YLG foraging strategies. I can see where you are going with this, it just lacks a clear flow.

Line 229: add ‘s’ to ‘regulation’

Line 234: Not sure if I mentioned it in previous revision, I personally don’t think figures should be mentioned in a discussion if they are properly interpreted in the results.

Line 240: ‘…could be the result of changes in foraging strategies and behaviour…’

Line 245: Suggestion: Although evidence of increase diet overlap between the studied species is lacking…

Line 253: A second explanation for what?

Line 253-259: You described the drivers of breeding success for shearwaters but not for gulls. What would be the difference between shearwater and gulls that could explain the synchrony? or environmental buffering (see comment line 253).

Line 262-263: Say what was the change in population size (increase or decrease).

Line 265-268: I suggest splitting and rewriting the sentence like this: However, we argue that the mechanisms from transient dynamic do not explain the disappearance of synchrony after seven years, and that changes in interspecific competition at sea is likely the cause of this pattern.

Typos and double spaces: Lines 231; 238; 240; 246; 254; 261; 262; 266

Figure 2: Add the value of R2. Y-axis put ‘Annual mean egg volume per clutch’

I’m curious whether you tested the leverage of the data at WNao -5. I’m concerned that it’s driving the slope of your relationship, and it wouldn’t be significant otherwise.

7. PLOS authors have the option to publish the peer review history of their article (what does this mean?). If published, this will include your full peer review and any attached files.

Reviewer #1: **Yes: **Alejandro Sotillo

Reviewer #2: No

---

## [Author Response · Author response to Decision Letter 1]

16 Sep 2022

I would like to congratulate the authors for their hard work on improving this manuscript and addressing the reviewer’s comments. Most of my concerns were address by simply clarifying the method which was greatly appreciated. The paper has improved significantly, and I only have minor suggestions. 

#Authors:

Dear editor, 

We would like to thank you and the reviewers for the time you have dedicated to review our manuscript. We believe that your suggestions have greatly improved the quality of our manuscript. Finally, we would like to thank you for considering our manuscript for publication in PLOS ONE. 

On behalf of all authors, 

Payo-Payo, A 

L44: add ‘a’ before pronounced.

#Authors: We have added ‘a’ before pronounced as suggested. 

L50-51: Sentence is unclear. Suggestion: Intra-specific synchrony is apparent a at large spatial scale from correlated environmental conditions, the so-called ‘Moran’s effect’. I’m not sure if this is what you were trying to say. 

#Authors: We have rephrased as suggested “Intra-specific synchrony is apparent a at large spatial scale from correlated environmental conditions, the so-called ‘Moran’s effect”.

L63-65: I’m having trouble connecting the start and end of this sentence. I suggest making it into 2 sentences: Breeding parameter – such as egg volume- often reflect conditions in the ecosystems. Therefore, using inter-specific synchrony of breeding parameters could be a useful indicator of local environment state.

#Authors: We have rephrased as suggested “Breeding parameter – such as egg volume- often reflect conditions in the ecosystems. Therefore, using inter-specific synchrony of breeding parameters could be a useful indicator of local environment state.”

Line 67: Your discussion now clearly answers your hypothesis, but I would mention your subsidy-decoupling hypothesis somewhere in your discussion.

#Authors: We have now included the hypothesis in the discussion. 

Line 83-84: I don’t think a subsection is necessary here.

#Authors: We have removed subsection heading as suggested. 

Line 93-123: This paragraph is very long. I suggest merging line 93-96 to previous paragraph and splitting the paragraph by species for the remaining lines (splitting somewhere around 108).

#Authors: We have redistributed paragraphs as suggested. 

Line 108: Does YLG always breed close by to shearwaters or is that inly in the Mallorca region?

#Authors: We have rephrased to “Yellow-legged gulls at our studied colony breed sympatrically” 

Line 121-123: I appreciate the addition; however, I’m still craving for more detail. Suggestion: add ‘so that egg volume is higher when environmental stochasticity is removed’ at the end of the sentence.

#Authors: We have added “so that egg volume is larger when environmental stochasticity is removed.” as suggested. 

Line 137-138: Thank you for clarifying that you calculated the mean of egg volume per clutch prior to calculating the annual mean. Maybe mean egg volume per clutch instead? Was the annual mean used for both the synchrony and the GLM? At line 190-191, you only refer to ‘mean egg volume’ which suggest to me that you used mean egg volume per clutch and not the annual mean (which is appropriate for the GLM). If so, the annual mean should only be mentioned in the synchrony part. If not correct line 190-191. 

#Authors: Apologies, yes that what we meant. We have now corrected line 137-138 as required. 

Line 204: Add a sentence precising that synchrony stopped after 2017. This is a big part of the discussion but was not apparent to me until then. 

#Authors: We have now included “Our results show that synchrony stopped after 2017.” as suggested by the editor. 

Line 222: Correlation? Why not the actual relationship with you B estimate? Do you have an actual value of R2 to add to the figure? 

#Authors: We have now added the R2 and the βWNAO value to the figure caption. 

Line 228-252: Long paragraph and a little confusing. I would first split at line 236. Then I would reshuffle the sentences around: all sentence mentioning overlap first, then YLG foraging strategies. I can see where you are going with this, it just lacks a clear flow. 

#Authors: We have split, and reorder as requested. 

Line 229: add ‘s’ to ‘regulation’

#Authors: We have added ‘s’ to ‘regulation’ as suggested. 

Line 234: Not sure if I mentioned it in previous revision, I personally don’t think figures should be mentioned in a discussion if they are properly interpreted in the results. 

#Authors: We have removed references to figures in the discussion as suggested by the editor. 

Line 240: ‘…could be the result of changes in foraging strategies and behaviour…’

#Authors: We have rephrased as suggested. 

Line 245: Suggestion: Although evidence of increase diet overlap between the studied species is lacking…

#Authors: We have rephrased as suggested

Line 253: A second explanation for what?

#Authors: We have included “explanation for the observed synchronization pattern” to clarify our statement. 

Line 253-259: You described the drivers of breeding success for shearwaters but not for gulls. What would be the difference between shearwater and gulls that could explain the synchrony? or environmental buffering (see comment line 253).

#Authors: We have now included the following sentences following the editors suggestion “ We know the egg volume and clutch size of Yellow-legged gulls in our population during the 7-year synchrony period was driven by food waste availability and that they started consuming a more marine diet after the landfill was closed (Payo-Payo et al., 2016). It is possible that marine prey exploited by Yellow-legged gull and Scopoli’s Shearwater are susceptible to similar environmental forcing and results in coupled dynamics.” 

Line 262-263: Say what was the change in population size (increase or decrease).

#Authors: We have specified that the change was a decrease. 

Line 265-268: I suggest splitting and rewriting the sentence like this: However, we argue that the mechanisms from transient dynamic do not explain the disappearance of synchrony after seven years, and that changes in interspecific competition at sea is likely the cause of this pattern. 

#Authors: We have rephrased as suggested by the editor. 

Typos and double spaces: Lines 231; 238; 240; 246; 254; 261; 262; 266

#Authors: We have removed double spaces. 

Figure 2: Add the value of R2. Y-axis put ‘Annual mean egg volume per clutch’

#Authors: We have included the value of R2 in the figure caption and amended the Y-axis as suggested. 

I’m curious whether you tested the leverage of the data at WNao -5. I’m concerned that it’s driving the slope of your relationship, and it wouldn’t be significant otherwise.

#Authors: We had not done it before the editor suggested it. But we have now tested it and the correlation between WNAO-5 and Egg volume is non-significant with R2= 0.011, βwnao=0.08 ±0.606. 

We thank the editor and the reviewers for their constructive comments, and we hope our responses satisfy their minor comments.

---

## [Editor Report · Decision Letter 2]

20 Sep 2022

Interspecific synchrony on breeding performance and the role of anthropogenic food subsidies.

PONE-D-22-11852R2

Dear Dr. Payo Payo,

We’re pleased to inform you that your manuscript has been judged scientifically suitable for publication and will be formally accepted for publication once it meets all outstanding technical requirements.

Kind regards,

Jose M. Riascos, Ph.D.

Academic Editor

PLOS ONE
---

## [Editor Report · Acceptance letter]

22 Sep 2022

PONE-D-22-11852R2 

Interspecific synchrony on breeding performance and the role of anthropogenic food subsidies. 

Dear Dr. Payo-Payo:

I'm pleased to inform you that your manuscript has been deemed suitable for publication in PLOS ONE. Congratulations! Your manuscript is now with our production department. 

Kind regards, 

on behalf of

Professor Jose M. Riascos 

Academic Editor

PLOS ONE